# Prostatic hyperplasia: Vascularization, hemodynamic and hormonal analysis of dogs treated with finasteride or orchiectomy

**Daniel S. R. Angrimani, Maria Claudia P. Francischini, Maíra M. Brito, Camila I. Vannucchi** *

Department of Animal Reproduction, Faculty of Veterinary Medicine, University of São Paulo, São Paulo, Brazil

* cacavann@usp.br

**Data Availability Statement:** All relevant data are within the manuscript.

**Funding:** CIV granted by São Paulo Research Foundation (http://www.fapesp.br/en/) - FAPESP

## Abstract

As a consequence of a hormonal imbalance, Prostatic Hyperplasia (PH) is characterized by increased prostate volume, along with higher local angiogenesis and vascularization. Orchiectomy is the common treatment for dogs, however it is not an option for breeding animals. Thus, finasteride arises as the drug of choice for stud dogs. Therefore, the aim of this study was to evaluate the effects of orchiectomy or finasteride therapies on hormonal and vascular dynamics of PH dogs. Fifteen dogs, aged 6–13 years were assigned to: Untreated Group (dogs diagnosed with PH—n = 5), Finasteride treated group (PH dogs treated with finasteride—n = 5) and Orchiectomy treated group (PH dogs submitted to orchiectomy–n = 5). Evaluations were performed in a monthly interval (first day of treatment; after 30 and 60 days). Doppler ultrasonography was performed to measure prostatic volume, vascularization and hemodynamic profile of prostatic artery. Dihydrotestosterone, estrogen and testosterone concentrations were measured. At day 60, prostatic biopsy was performed for histological, immunohistochemical and qPCR analysis for *VEGF-A* expression. At day 60, vascularization score was higher in untreated compared to treated groups (finasteride and orchiectomy). Furthermore, *VEGF-A* expression was lower in the Orchiectomy Treated Group, but VEGF-A was immunohistochemically lower in both treated groups (finasteride and orchiectomy) compared to the Untreated Group. The efficiency of finasteride treatment in reducing clinical signs, prostate volume and vascularization appears to be similar to orchiectomy. In conclusion, both PH medical and surgical therapy lead to reduction in prostate dimension and *VEGF-A* expression and, consequently, lower local vascularization. However, orchiectomy promotes marked hormonal changes, which ultimately lead to prostate atrophy.

## 1. Introduction

Prostatic Hyperplasia (PH) is the most common male disease of canine senescence [1], with incidence of 95% in non-castrated dogs aged 9 years and 50% and 16%, respectively, in those

2015/05419-5 DSRA granted by São Paulo
Research Foundation (http://www.fapesp.br/en/) -
FAPESP 2013/25966-5 The funders had no role in
study design, data collection and analysis, decision
to publish, or preparation of the manuscript.

aged 5 and 2 years [2]. The most common observed clinical signs of PH in dogs are tenesmus, dysuria, hematuria and hematospermia [3]. However, symptomatic PH occurs when the increase in prostate volume compresses urinary and intestinal segments [3]. Canine PH has high similarity with the disease in elderly men, featuring the dog as a good animal model for the study on humans [4].

PH is triggered by a specific hormonal imbalance of aged dogs and men, characterized by 40% less testosterone and 60% more estradiol concentration compared to young individuals [5]. In addition, there is increase in prostatic conversion of testosterone into dihydrotestosterone (DHT), catalyzed by the 5-alpha reductase enzyme [6]. High DHT concentration leads to increased production and expression of prostate growth factors (e.g. vascular endothelial growth factor [VEGF]), giving rise to hyperplasia of the prostatic gland [6].

The definitive treatment for PH in dogs is bilateral orchiectomy, which aims to cease testosterone production and, ultimately, reduce its conversion into DHT [7]. Such endocrine change leads to lower expression of prostate VEGF, resulting in reduced blood flow to the prostatic parenchyma [8]. However, gonadectomy is not an option for stud dogs of high genetic value [7] or PH-affected men [9]. Although no medical therapy for PH is effective such as orchiectomy in reducing clinical signs and prostatic size [10], conservative treatment should to be considered for special-case patients. Estrogen therapy has been initially used for such purpose, aiming to block luteinizing hormone release and, consequently, testosterone synthesis by Leydig cells [11]. However, estrogen treatment is not considered fully safe due to its toxic potential and side effects in males [12]. Thus, GnRH analogs or antagonists have been considered choices for PH treatment, while indirectly reducing testosterone concentrations [13]. Although effective in decreasing prostate size, blood flow and hormone levels, GnRH treatment interferes with testicular sperm production and male fertility, which is not an advantage comparing to gonadectomy in dogs [14]. In this context, PH therapy with finasteride is an alternative for decreasing DHT concentration by 5α-reductase inhibition, reducing prostate PH volume and clinical signs without altering the seminal quality of dogs [15,16]. In addition, finasteride is the drug of choice for PH treatment in men [17]. It has been previously shown that finasteride was capable of reducing prostate volume and local vascularization after 60 days of treatment in dogs [18]. However, the effect of finasteride on tissue expression of vascular factors has not been deeply explored both in humans and dogs [19].

Therefore, this study aimed to characterize the expression of prostatic VEGF, hemodynamic and blood flow changes of the prostatic artery and hormonal PH levels of dogs submitted to finasteride orchiectomy treatment.

## 2. Materials and methods

### 2.1. Animals and experimental study

The Bioethics Committee of the School of Veterinary Medicine and Animal Science—University of São Paulo (protocol number 7122171213) approved the current study. Privately owned dogs of several breeds were used in this study, and the animals were included only after owner consent.

Fifteen dogs of several breeds, with body weights ranging from 10–30 kg and aged 6 to 13 years were presumptively diagnosed with PH through clinical signs of hematospermia, tenesmus, dysuria and hematuria, as well as prostatic biometry by B-mode ultrasonography [18,20,21]. Dogs were randomly assigned to 3 experimental groups, according to the treatment mode:

- Untreated Group (n = 5);

- Finasteride treated Group (n = 5): dogs treated with 5 mg/per animal/day of finasteride (Finasterida®/Medley®) [22];

- Orchiectomy treated Group (n = 5).

To assure the appropriate sample size, an analysis was conducted with the SAS Power and Sample Size 12 (SAS Institute Inc., Cary, NC, USA). A retrospective analysis of the variables indicated there was a power of 0.99, which is considered an acceptable statistical power (at least 0.8). Hence, 5 males per group were sufficient to demonstrate significant differences in the data.

Doppler ultrasonography and hormonal evaluations were performed in a monthly interval during 60 days, considering the first day of the diagnosis, medical treatment or gonadectomy. A close clinical follow-up was performed with special attention to possible adverse effects and remission of clinical signs. After 2 months of treatment or diagnosis, dogs were subjected to prostatic biopsy in order to obtain prostate fragments for histological analysis, immunohisto-chemical and qPCR *VEGF-A* expression. Under general anesthesia, dogs were placed in dorsal recumbence and a parapenile small incision was performed. An 18-gauge semi-automatic Tru-Cut needle was inserted through the incision and, by means of an ultrasound-guide, prostatic tissue samples were collected through at least two punctures of one of the prostatic lobes [23,24].

The fragments were submitted to real-time PCR processing or immediately washed in 0.9% saline solution, fixed in 10% buffered formaldehyde phosphate solution for 24 hours and posteriorly stored in 70% alcohol until inclusion in paraffin.

## 2.2. B-mode and doppler prostate ultrasonography

Prostatic ultrasonographic assessment was performed transabdominally with a linear 5.5 MHz transducer (M5 Mindray®, Shenzhen, China). Dogs were placed in dorsal recumbency and caudal abdominal region was sheaved before transabdominal scanning.

Prostatic volume (PV) was evaluated by B-mode ultrasonography using the bladder as a window, measuring height and length in the sagittal plane and width in the axial plane. PV was calculated using the formula: PV (cm$^3$) = 0.487 × L × W × (DL+DT):2 + 6.38 (L = length; DL = depth on longitudinal section; DT = depth on transverse section; W = width) [25]. The expected prostate volume (EPV) was used to estimate the prostatic volume according to the dog weight, i.e., a prostatic volume control for each dog bodyweight, considering the following formula: EPV = 8.48 + (0.238 × Kg body weight) [20,25].

Prostatic tissue perfusion and blood flow velocity of the prostatic artery were evaluated by Doppler ultrasonography. Prostatic artery was scanned at the hypogastric abdominal region and located cranio-dorsal to the prostate gland [20,26]. Color flow Doppler was used to map the vessel and subsequently pulsed-wave Doppler was used to characterize the waveform. Blood sample volume was positioned at the artery center and all measurements were obtained with an angle of ≤60˚, making proper angle correction whenever necessary. A total of 9 stable waves of the prostatic artery were obtained to calculate the average of each variable. The following blood flow velocity parameters were automatically calculated by the Doppler machine software, using mathematical formulas or Pourcelot index: peak systolic velocity (PSV), end diastolic velocity (EDV), resistance index [RI = (PSV - EDV) / PSV], pulsatility index [PI = (PSV - EDV) / mean velocity], time average maximum velocity (TAMAX) and peak systolic: diastolic velocity [S/D = (PSV / EDV)].

For the qualitative evaluation of the prostate tissue perfusion, prostate scanning was preferably carried out in longitudinal section by B-mode ultrasonography, followed by color flow

Doppler. Prostate vascularization was graded on an arbitrary scale of 1–3, being 1 –the minimum degree of prostate vasculature; 2 –the intermediary degree of vascularization and 3 –the maximum degree of prostate vasculature. Analysis was performed always by only one analyzer.

## 2.3. Hormonal profile of testosterone, estrogen and dihydrotestosterone

Blood samples (3 mL) from all dogs were collected by puncture of the right or left cephalic or jugular veins into vacuum tubes without an anticoagulant at monthly intervals during the morning period and allowed to clot at room temperature. The tubes were then centrifuged for 10 minutes at $1500 \times g$, and serum was separated in aliquots of 0.5 mL and stored at −20°C until hormone analysis.

Testosterone and estrogen concentration was measured using a commercial radiomunoassay kit (BECKMAN Coulter Company IM-1841 / IM-1119-Immunotech, Prague, Czech Republic and BECKMAN Coulter Company IM-1841 / DSL4800-Immunotech, Prague, Czech Republic, respectively), both previously validated for dogs [13,27] For dihydrotestosterone, competitive ELISA kit (BECKMAN, Active® DSL9600i-Immunotech-, Prague, Czech Republic) was used [28]. The limit detection for the testosterone assay was 0.01 ng/mL and the intra-assay coefficient of variation was 4.52%. For estrogen, the limit detection test was 1.12 pg/mL and the coefficient of variation ranged from 7.15% to 3.45%. For dihydrotestosterone assay, the limit detection was 6.58% and the coefficient of variation ranged from 9.86% to 3.03%.

## 2.4. Histological evaluation of prostatic fragments

For the evaluation of the histo-morphological appearance of the prostate tissue, 5 μm histological sections were dewaxed and stained with hematoxylin-eosin (HE) for analysis under light microscopy (Nikon, Eclipse E200, Japan). PH was considered to occur according to the classification of Palmieri, et al. [29]. In other words, PH consisted of normal prostate architecture, whilst an increased amount of secretory epithelium. The alveoli region was expected to be larger and presenting papillary projections, as well as dilated and cystic, lined by columnar to small cuboidal cells with absence of secretory activity. The atrophic areas had mild to moderate increased stroma.

## 2.5. Immunohistochemical reaction for vascular endothelial growth factor (VEGF-A)

For the immunohistochemical analysis of VEGF-A, 3 μm prostate sections were used. Antigen retrieval was performed in Tris/EDTA buffer at pH 9 with the primary antibody and subsequently with the secondary antibody. A mouse monoclonal antiserum specific for VEGF-A (VG-1, AB1316, ABCAM, UK; diluted 1 in 100) was used in an overnight system. Subsequently, an incubation with antirabbit IgG secondary antibody (STAR124P, Bio-Rad Laboratories, USA; diluted 1:1000) was performed, followed by the addition of 3,3'-diaminobenzidine substrate (during 5 minutes) and haematoxylin counterstain [30].

All immunohistochemical reactions were accompanied by a negative control as previously described [31, 32], consisted of prostate tissue from the Untreated Group lacking application of primary antibody. For the positive control, we used canine mammary gland tumor, previously attested by others [33–35] and also as a guarantee of the cross-reactivity of the primary antibody with canine tissue. Evaluations were performed by microscopy (Leitz, Dialux 20, Germany) at 400x magnification by three operators in a single-blind manner (results were calculated as the mean value of three evaluations). Only the cytoplasm of the prostate glandular epithelium within ten fields was analyzed, by using the relationship between the percentage of

stained area (0–0 to 5%, 1–5 to 33%; 2–33 to 66%; 3—above 66%) and intensity (0—negative, 1—low, 2—moderate, 3—high). Thus, a score was defined by the following calculation: Score = Percentage of stained area $x$ staining intensity [36].

## 2.6. Evaluation of prostatic vascular endothelial growth factor (VEGF-A) by quantitative PCR

After biopsy, prostate fragments were transferred to cryotubes, immersed in liquid nitrogen and stored in -80˚C for subsequent RNA extraction. Samples were macerated and RNA extraction was performed using the RNAspin mini Illustrative extraction kit (GE Healthcare Life Sciences, Piscataway, NJ). After extraction, the RNA was quantified by NanoDrop spectrophotometer (Thermo Fisher Scientific, Massachusetts, USA), using 1 µL of the sample. Subsequently, 1 µg of total RNA was used for cDNA synthesis, through the commercial Superscript VILO cDNA Synthesis kit (Life Technologies). Then, synthesized cDNA was quantified using the Qubit 2.0 Fluorometer (Qubit dsDNA BR, Life Techonologies) and samples were stored at -20˚C.

To determine the gene expression of vascular endothelial growth factor (*VEGF-A*), the primer was selected according to Giantin, et al. [30] (Forward primer: 5'– CGTGCCCACT GAGGAGTT–3'/Reverse primer: 5'– GCCTTGATGAGGTTTGATCC–3'). The primers of the endogenous genes B2M (Forward primer: 5'–TCCTCATCCTCCTCGCT–3'/Reverse primer: 5'–TTCTCTGCTGGGTGTCG–3') and GAPDH (Forward primer: 5'–GTAGTG AAGCAGGCATCGGA–3'/Reverse primer: 5'–GTCGAAGGTGGAAGAGTGGG–3') were designed using online primer-Blast software (http://www.ncbi.nlm.nih.gov/tools/primer-blast/index.cgi?LINK_LOC=BlastHome, NCBI, NIH, USA). The standardization of the primers was performed through a standard serial dilution curve of samples 1:10 with 7 points.

Reactions were performed in triplicates for each sample in Realplex2 Mastercycler (Eppendorf, Hamburg, Germany). For each reaction, the SYBR Greener kit qPCR Supermix Universal (Life Technologies) and 2 µL of the cDNA were used. The qPCR program consisted of 2 minutes at 50˚C, 10 minutes at 95˚C and 40 cycles of 95˚C for 15 seconds and 60˚C for 60 seconds. The concentration of the primers was defined as 200:200 for *VEGF-A* and GAPDH and 300:300 for B2M.

The efficiency of the reaction was calculated by the PCR software Realplex, with ideal efficiency considered between 0.9 and 1.1. The amplification of each gene in each sample was evaluated to verify if Ct values were contained in the linear standard curve. The qRT-PCR data were analyzed using the ΔΔCt method [37]. The geometric mean of housekeeping genes GAPDH and B2M were used to normalize the relative quantification of target genes. The Ct value of this sample was then subtracted from the other samples to generate the value of ΔΔCt and consequently the Fold Change of gene expression.

## 2.7. Statistical analysis

Data were evaluated using SAS System for Windows (SAS Institute Inc., Cary, NC, USA). Effects of experimental group, time of evaluation, and interaction between these factors were estimated by repeated measures ANOVA (SAS MIXED procedure) for the ultrasonographic parameters and hormonal profile. Differences between treatments were analyzed by means of parametric and non-parametric tests, according to the residual normality (Gaussian distribution) and homogeneity of variance. Whenever one of these assumptions has not been complied, the data was transformed. If transformations were not successful, non-parametric tests were used. Testosterone concentration, estrogen concentration, dihydrotestosterone concentration, pulsatility index, peak systolic:diastolic velocity and vascularization score were log

transformed. In the absence of significant interactions, the effect of groups were analyzed by merging all the evaluation moments, and conversely, evaluation times were compared by combining all groups, taking into account the Bonferroni correction (PROC GLIMMIX). Otherwise, comparisons were made taking into account the main effects.

Differences between groups were analyzed by an orthogonal contrast. Orthogonal comparisons were performed to determine the two main effects: effect of treatment (Untreated Group *vs*. Finasteride Treated Group + Orchiectomy Treated Group) and the effect of both treatments (Finasteride Treated Group *vs*. Orchiectomy Treated Group). The Least Significant Difference (LSD) test was used to compare moments of evaluation within the experimental period.

For prostatic volume (PV) and expected prostatic volume (EPV), results were analyzed comparing all three experimental groups (Untreated, Finasteride Treated and Orchiectomy Treated groups) using LSD test in each moment of evaluation (Day 0, 30 and 60). Moreover, in order to compare PV and EPV, Student t test (parametric variables) and Wilcoxon test (nonparametric variables) were used.

The relative qPCR data were analyzed by the MIXED procedure, as described by Steibel, et al. [38]. The response variables were also submitted to Spearman correlation analysis. Results are reported as untransformed means ± S.E.M. Level of significance was P<0.05.

## 3. Results

Significant interactions (*P<0.05*) between groups (Untreated *vs*. Finasteride Treated *vs*. Orchiectomy Treated groups) and moments (Days 0 *vs*. 30 *vs*. 60) were observed for prostate volume, vascularization score, peak systolic: diastolic velocity (S/D), resistance index (RI) and testosterone and dihydrotestosterone concentrations (Table 1).

After 30 days of the onset of the experiment, all dogs of the Finasteride Treated and the Orchiectomy Treated groups no longer presented the initial clinical signs (hematospermia, tenesmus, dysuria or hematuria). Moreover, dogs of the Untreated Group remained clinically stable, allowing the continuity of the experimental design.

### 3.1. B-mode and doppler ultrasonographic analysis

Prostate volume was higher than EPV on day 0 and day 30 in all experimental groups (Fig 1). However, no difference between PV and EPV occurred in day 60 for the Finasteride Treated

**Table 1. Probability values for the interaction between the main effects of treatment (Untreated *vs*. Finasteride Treated *vs*. Orchiectomy Treated) and time (0 *vs*. 30 *vs*. 60 days) on ultrasonographic analysis and hormonal concentrations.**

|  | Treatment | Time | Treatment x Time |
|---|---|---|---|
| PVS (cm/s) | 0.44 | 0.42 | 0.47 |
| EDV (cm/s) | 0.13 | 0.16 | 0.33 |
| TAMAX (cm/s) | 0.32 | 0.33 | 0.56 |
| **S/D** | **0.03** | **0.02** | **0.01** |
| **Resistance index (RI)** | **0.08** | **0.15** | **0.02** |
| Pulsatility index (PI) | 0.43 | 0.98 | 0.97 |
| **Prostatic volume (cm³)** | **0.20** | **<0.01** | **<0.01** |
| **Vascularization score (1–3)** | **<0.01** | **<0.01** | **0.03** |
| Estrogen (pg/mL) | 0.30 | 0.02 | 0.72 |
| **Testosterone (ng/mL)** | **<0.01** | **<0.01** | **<0.01** |
| **Dihydrotestosterone (pg/mL)** | **<0.01** | **<0.01** | **0.01** |

PSV: Peak systolic velocity; EDV: End diastolic velocity; TAMAX: Time average maximum velocity; S/D: Peak systolic: diastolic velocity.

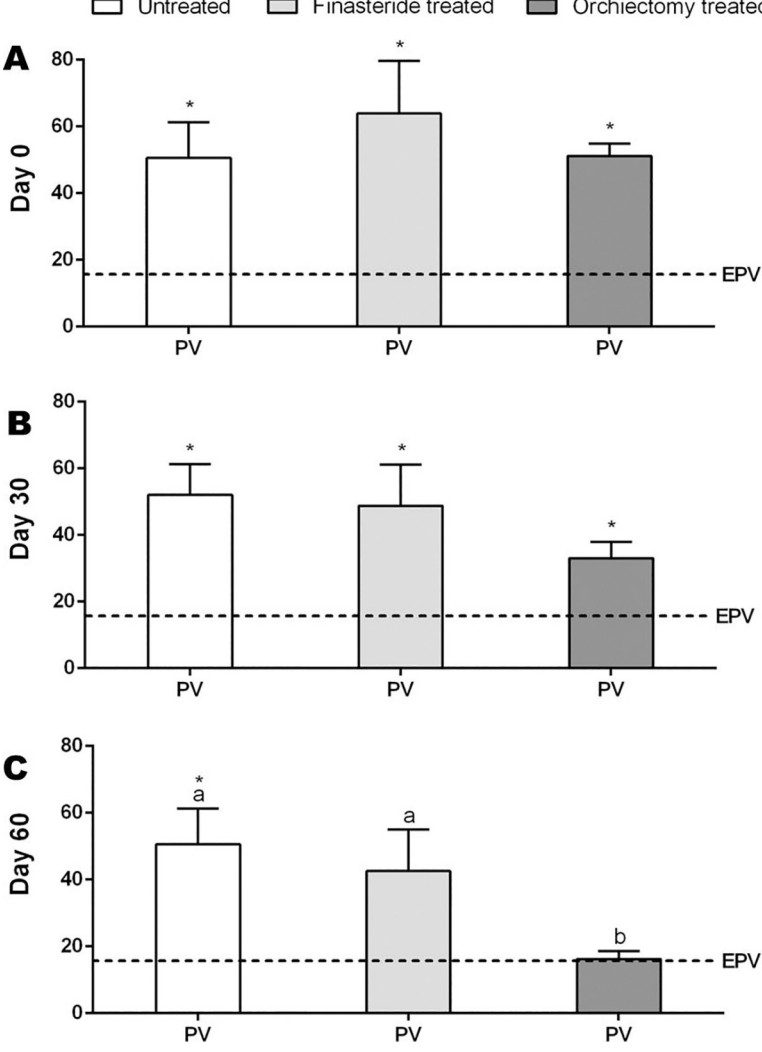

**Fig 1.** Prostate Volume (PV–cm$^3$) and Expected Prostate Volume (EPV–cm$^3$) in the Untreated, Finasteride Treated and Orchiectomy Treated groups at day 0 (A), 30 (B) and 60 (C). *indicate significant differences between PV and EPV and $^{a,b}$ indicate significant differences between groups (P<0.05).

Group (PV: 42.5±12.3 cm$^3$; EPV: 14.3±0.6 cm$^3$) and the Orchiectomy Treated Group (PV: 16.2±2.4 cm$^3$; EPV: 15.13±0.3 cm$^3$) (Fig 1). After 60 days, prostate volume of the Orchiectomy Treated Group was lower than the other groups (Fig 1). Moreover, a prostate volume reduction of 67.4±6.1% was observed in orchiectomized dogs, different from those treated with finasteride (35.2±5.4%) and untreated (15.4±7.9% increase in prostate volume).

The Orchiectomy Treated Group had significant increase in peak systolic: diastolic velocity (S/D) of the prostatic artery at 60 days, compared to the Finasteride Treated Group (Fig 2A). However, at this time-point, the Untreated Group had lower values of S/D compared to both treatment groups. In addition, S/D was lower in the Untreated Group on day 30, compared to the other groups (Fig 2A). Significant increase in resistance index of the prostatic artery was observed over time for the Orchiectomy Treated Group (Fig 2B). Moreover, on day 60, RI was lower in the Untreated Group compared to both treatment groups (Fig 2B). In the Finasteride and Orchiectomy Treated groups, there was a significant reduction in vascularization score over time (Fig 2C). On days 30 and 60, the Untreated Group had the highest vascularization

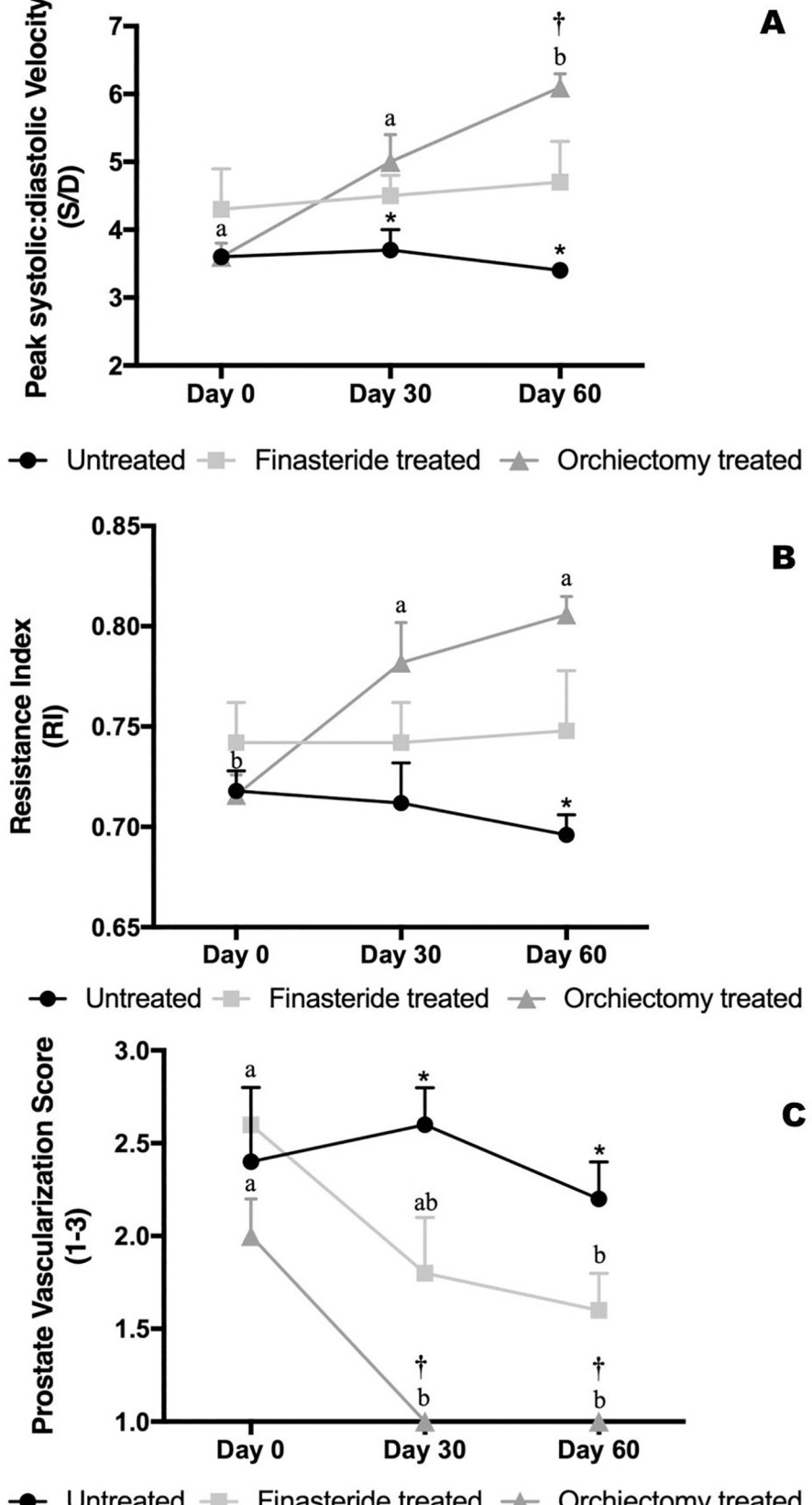

**Fig 2.** Peak systolic: diastolic velocity (A) and resistance index (B) of the prostatic artery and prostate vascularization score (C) in the Untreated, Finasteride Treated and Orchiectomy Treated groups throughout the experimental period. *indicate: Contrast 1 –comparison between Untreated *vs*. Treatments (Finasteride and Orchiectomy) at P<0.0001. †indicate: Contrast 2 –comparison between Finasteride *vs*. Orchiectomy Treated groups at P<0.05.

score compared to both the Finasteride and Orchiectomy Treated groups (Fig 2C). Furthermore, on day 30 and 60, the Orchiectomy Treated Group had lower prostate vascularization score compared to the Finasteride Treated Group (Fig 2C).

The Untreated Group had higher end diastolic velocity (EDV) compared to both treatment groups. In addition, EDV of the Finasteride Treated Group was higher than Orchiectomy Treated Group (Table 2). Time average maximum velocity (TAMAX) was lower in the Orchiectomy Treated Group compared to the Finasteride Treated Group (Table 2).

### 3.2. Hormonal profile

No significant difference between groups or time-points was verified for estrogen concentrations (Table 2). On the other hand, a significant reduction in testosterone concentrations was observed in the Orchiectomy Treated Group throughout the experiment (Fig 3A). The Untreated Group had higher testosterone concentrations compared to both treatment groups (Fig 3A), however, the Orchiectomy Treated Group had lower testosterone concentrations than the Finasteride Treated Group on days 30 and 60 (Fig 3A).

After 30 and 60 days, dihydrotestosterone concentrations decreased significantly in the Orchiectomy Treated Group, compared to the Finasteride treated Group, although the latest group had gradual decrease in DHT concentrations (Fig 3B). The Untreated Group had higher DHT concentrations on day 60 compared to both treatment groups (Fig 3B).

### 3.3. Histological analysis

The Untreated and Finasteride Treated groups had similar histological pattern, with hypertrophy of the prostate glandular epithelium and increased space between glands, glandular proliferation, papillary projection and mild stromal fibrosis. Several alveoli were dilated and cystic, presenting stasis of secretory activity. Inflammatory cells were observed in both groups.

On the other hand, the Orchiectomy Treated Group presented massive reduction of the glandular epithelium, remaining only connective tissue, characterizing an atrophic prostate gland.

**Table 2. Mean ± SE of prostate ultrasonographic parameters and estrogen concentrations of Untreated, Finasteride Treated and Orchiectomy Treated groups and Probability values (P) of statistical orthogonal contrast analysis.**

| | Groups | | | Contrast | |
| --- | --- | --- | --- | --- | --- |
| | **Untreated** | **Finasteride Treated** | **Orchiectomy Treated** | **C1** | **C2** |
| PSV (cm/s) | 71.2±2.5 | 73.3±4.1 | 64.3±4 | 0.6 | 0.08 |
| EDV(cm/s) | 20.3±0.78 | 18.8±1.5 | 14.9±1.3 | **0.03** | **0.03** |
| TAMAX (cm/s) | 26±1.90 | 28.7±2.1 | 22.4±1.9 | 0.8 | **0.03** |
| Pulsatility index | 2.1±0.2 | 1.9±0.1 | 2.5±0.2 | 0.5 | 0.08 |
| Estrogen (pg/mL) | 6.74±2.1 | 3.62±0.7 | 2.99±0.5 | 0.1 | 0.6 |

Contrast C1 –Comparison between Untreated and Treated Groups (Finasteride Treated Group + Orchiectomy Treated Group); Contrast C2 –Comparison between Finasteride Treated Group and Orchiectomy Treated Group; PSV: Peak systolic velocity; EDV: End diastolic velocity; TAMAX: Time average maximum velocity; S/D: Peak systolic: diastolic velocity.

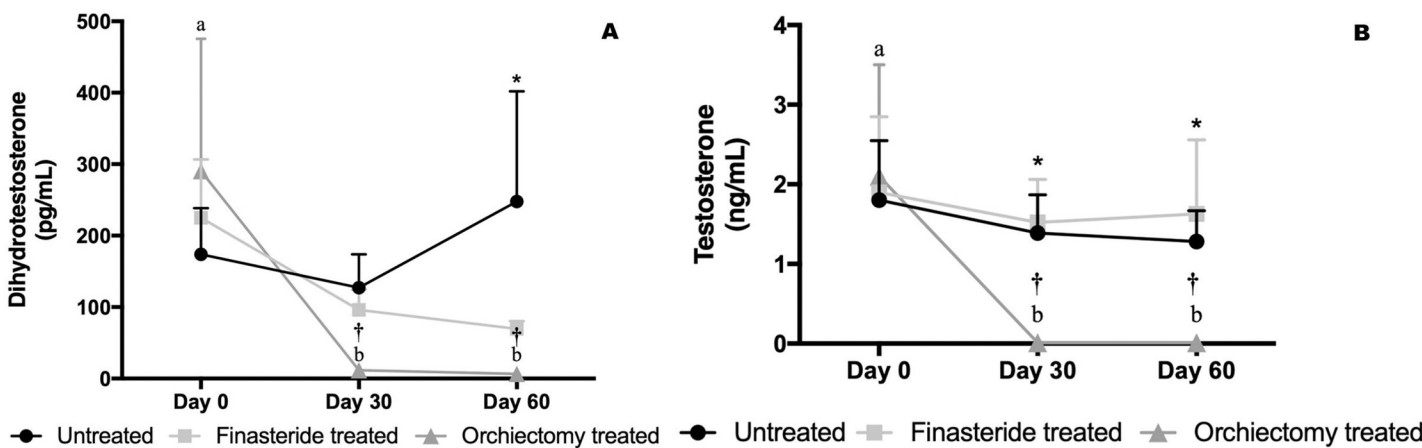

**Fig 3.** Testosterone (A) and dihydrotestosterone (B) concentrations in the Untreated, Finasteride Treated and Orchiectomy Treated groups throughout the experimental period. *indicate: Contrast 1 –comparison between Untreated *vs*. Treatments (Finasteride and Orchiectomy) at P<0.01. †indicate: Contrast 2 – comparison between Finasteride *vs*. Orchiectomy Treated groups at P<0.0001.

### 3.4. Vascular endothelial growth factor analysis by immunohistochemical and qPCR

Regarding prostatic immunostaining score for VEGF-A, the Untreated Group had higher *VEGF-A* expression compared to both treatment groups (Fig 4). The Finasteride Treated

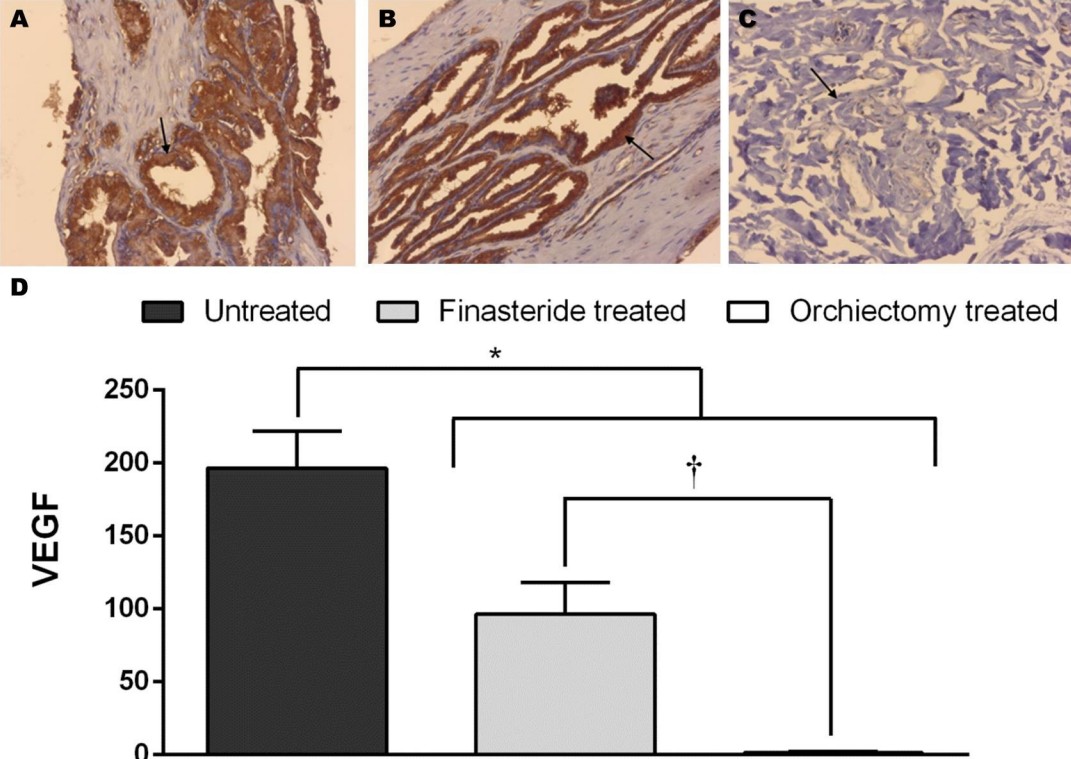

**Fig 4.** (I) Cytoplasmatic and glandular epithelium immunostaining for VEGF-A in the Untreated Group (arrow, A), Finasteride Treated Group (arrow, B) and Orchiectomy Group (arrow, C). 200x. (II) Mean and standard error of the immunostaining score for VEGF. *indicate: Contrast 1 –comparison between Untreated *vs*. Treatments (Finasteride and Orchiectomy) at P<0.0001. †indicate: Contrast 2 –comparison between Finasteride *vs*. Orchiectomy Treated groups at P<0.02.

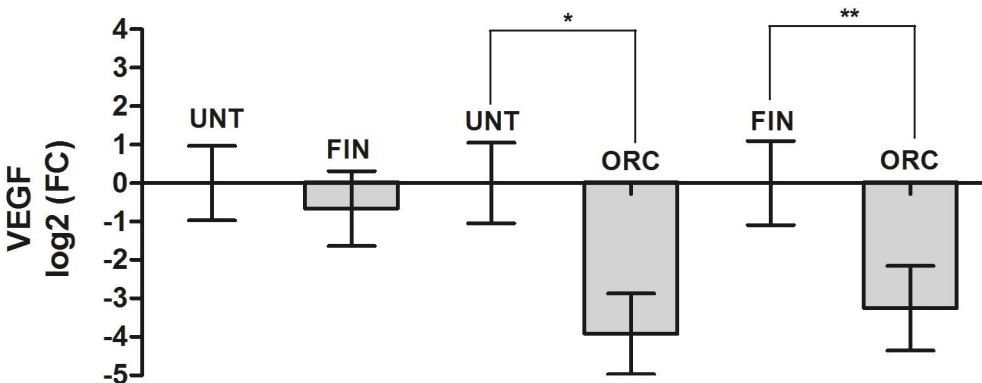

**Fig 5. Relative quantification of prostatic *VEGF-A* gene expression.** Values are expressed in log2 (FC, Fold Change). *indicate P<0.01. **indicate P<0.05. Legend: UNT–Untreated Group; FIN–Finasteride Treated Group; ORC–Orchiectomy Treated Group.

Group was superior than the Orchiectomy Treated Group (Fig 4). In addition, the Orchiectomy Treated Group had lower expression of *VEGF-A*, compared to the Untreated Group (P<0.004 –Fig 5) and the Finasteride Treated Group (P<0.01 –Fig 5).

### 3.5. Correlation analysis

Positive correlations were observed in the Finasteride Treated Group between prostate volume and vascularization score (r = 0.69; p = 0.0039). On the other hand, resistance index of the prostatic artery correlated negatively with prostate volume (r = -0.76; p = 0.0008) in the Orchiectomy Treated Group. In addition, there was a positive correlation between dihydrotestosterone concentrations and vascularization score (r = 0.85; p = 0.0002) in the Orchiectomy Treated Group.

## 4. Discussion

The present study aimed to evaluate possible changes in prostate vascularization and hemodynamic, as well as the hormonal profile of dogs submitted to medical or surgical treatment (finasteride or orchiectomy, respectively) for prostatic hyperplasia.

In PH dogs, an imbalance between testosterone and estrogen is observed [5], ultimately increasing dihydrotestosterone (DHT) concentrations [6]. In fact, in the present experiment, all animals presumptively diagnosed with PH (day 0) had characteristic hormonal profile (estrogen, DHT and testosterone) [39]. However, only orchiectomized dogs had significant decrease in DHT and testosterone concentrations. Surprisingly, the hormonal profile of the Finasteride Treated Group remained unchanged. However, it is noteworthy that men treated with finasteride present reduction of serum DHT concentrations only after 10 weeks of treatment [40]. Therefore, it was hypothesize that significant reductions in DHT concentrations may occur only after a long period of finasteride administration, since significant prostatic changes (atrophy of the prostate gland) were observed only after 9 weeks in dogs [41].

Finasteride treatment promotes reduction in prostatic size, reaching the expected prostatic volume (EPV) after 60 days of treatment, similarly to previous studies [18, 42,43]. Moreover, according to Iguer-Ouada; Verstegen [44], in our study, finasteride was effective in reducing PH clinical signs. On the other hand, dogs submitted to orchiectomy had remarkable prostate volume reduction, which can be attributed to the significant decrease in testosterone and DHT concentrations soon after gonadectomy. In fact, surgical treatment can reduce 50% the

prostatic volume by 3 weeks after orchiectomy [45], ultimately resulting in prostate atrophy [46]. In our experiment, orchiectomy led to prostatic atrophy 60 days after surgical procedure, reducing 67.4% the prostatic volume, as verified by the connective tissue content of the prostate glandular epithelium on histological analysis.

Untreated and finasteride-treated dogs had hypertrophy of the glandular tissue in a centrifugal manner, which is a peculiar histological finding of PH [29,47]. Among factors that stimulate mitosis of the prostatic parenchyma (e.g. epidermal factor, keratinocyte growth factor and insulin-like growth factors), VEGF-A is considered the main factor responsible for local angiogenesis [6,48]. Our data show that untreated PH dogs had the highest vascularization score throughout the experimental period, as well as the highest immunostaining score for VEGF-A in the prostate epithelium, similarly to previous data on men [49,50] and dogs [35,51,52]. Moreover, positive correlation between prostate volume and local blood flow was observed for untreated and finasteride-treated dogs, suggesting that increased prostatic size is followed by a concomitant increase in local vascularization in PH dogs [18,20,31].

Although we have not performed a tissue VEGF-A evaluation before treatment, a decrease in prostate vascularization score throughout the experimental period was observed in the Finasteride Treated and Orchiectomy Treated groups. This was attested by lower prostate immunostaining for VEGF-A and *VEGF-A* expression after 60 days of treatment, compared to untreated dogs. The most important modulator of the *VEGF-A* expression in the prostate tissue is the dihydrotestosterone concentrations [50]. Thus, reduction of DHT concentrations leads to lower *VEGF-A* expression, ultimately inhibiting neovascularization. In fact, a positive correlation between DHT concentrations and prostate vascularization score in the Orchiectomy Treated Group was verified, suggesting that the prostatic blood flow is under the control of the hormonal profile. The Orchiectomy Treated Group showed remarkable decrease in DHT and testosterone concentrations throughout the experimental period, and also significant reduction in prostate *VEGF-A* expression after surgery. Based on these results, it was suggested that severe androgen deprivation alters protein synthesis in a post-translational manner before the transcription of mRNA in prostatic tissues of dogs. Therefore, our data can be considered for further studies on prostate cancer and oncological treatments [51].

Despite the decrease in prostatic *VEGF-A* expression, local vascularization and prostate volume (35.2% reduction) in the Finasteride Treated Group, similarly to observed in men [52], hormonal change during the experimental period was not verified. Simultaneously to 5α-reductase blockage, finasteride also acts modulating the expression of prostatic androgen receptors [15,53]. Thus, it was hypothesized that the negative finasteride regulation of androgen receptors is the main causative factor for the reduction of prostatic volume during the initial period of treatment (while DHT concentrations have not yet changed), consequently, reducing tissue expression of *VEGF-A* and angiogenesis stimulus [53,54]. However, such assumption should be investigated in further studies regarding the local androgen receptor gene expression following long-term finasteride treatment. It could also be inferred that finasteride may also inhibit prostatic DHT receptors, thus blocking DHT hypertrophic action on prostate parenchyma. On the other hand, surgical treatment (orchiectomy) promotes intense prostate atrophy, leading to morpho-functional changes such as reduction of growth factors gene expression.

Gonadectomy is considered an important surgical treatment for the reduction of prostate volume, local vascularization and blood flow [53]. In our experiment, orchietomized dogs had lower tissue perfusion, as observed in ultrasonographic analysis. Thus, it was concluded that the sharp decrease in androgen profile (testosterone and DHT) after gonadectomy led to significant reduction in prostate blood flow. In fact, prostatic hemodynamics is primarily regulated by androgen hormones [55–60]. Thus, the expected reduction in prostate blood supply

after testicle removal is an indirect signal of treatment effectiveness [52]. In our experiment, negative correlation between prostate volume and resistance index of the prostatic artery in the Orchiectomy Treated Group was observed, suggesting that the increase in prostatic intravascular resistance can be used as a marker of prostatic size reduction and severe atrophy.

In conclusion, both PH medical and surgical therapy (finasteride and orchiectomy) led to reduction in prostate dimension and immunostaining for VEGF-A, consequently, lower local vascularization. However, orchiectomy promoted more marked androgenic decrease and severe prostate atrophy, in turn altering prostatic artery hemodynamic and even the *VEGF-A* expression. These results show a favorable prognosis for PH treated dogs and suggest further studies focused on the diagnosis and treatment of PH in dogs.

## Supporting information

**S1 Raw Data.**
(XLSX)

## Acknowledgments

The authors thank Dr. Camilla M Mendes, Dr. Thais R S Hamilton and Prof. Mayra E O A Assumpção for the qPCR analysis; Prof. Bruno Cogliati and Prof. Ricardo J G Pereira for the immunohistochemical evaluation; Prof. Claudio Alvarenga and Dr. Priscila Furtado for the hormonal evaluation and Dr. Gustavo Tiaen and Dr. Gisele Veiga for the prostatic biopsy and orchiectomy surgery, respectively.

## Author Contributions

**Conceptualization:** Daniel S. R. Angrimani, Camila I. Vannucchi.

**Data curation:** Daniel S. R. Angrimani.

**Formal analysis:** Daniel S. R. Angrimani.

**Funding acquisition:** Camila I. Vannucchi.

**Investigation:** Daniel S. R. Angrimani, Maria Claudia P. Francischini, Maíra M. Brito.

**Methodology:** Daniel S. R. Angrimani, Maria Claudia P. Francischini, Maíra M. Brito.

**Project administration:** Maria Claudia P. Francischini, Camila I. Vannucchi.

**Supervision:** Camila I. Vannucchi.

**Validation:** Maria Claudia P. Francischini.

**Writing – original draft:** Daniel S. R. Angrimani, Camila I. Vannucchi.

**Writing – review & editing:** Camila I. Vannucchi.

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
