## [Decision Letter · Decision Letter 0]

28 Apr 2020

PONE-D-20-09715

Benign Prostatic Hyperplasia: vascularization, hemodynamic and hormonal analysis of dogs treated with finasteride or orchiectomy

PLOS ONE

Dear Vannucchi

Thank you for submitting your manuscript to PLOS ONE. After careful consideration, we feel that it has merit and interest but does not fully meet PLOS ONE’s publication criteria as it currently stands. Therefore, we invite you to submit a revised version of the manuscript that addresses the points raised during the review process.

You will carefully address all the comments of the two reviewers. Especially, you will have to demonstrate the specificity of VEGF-A immunolabelling illustrated in Figure 5, by testing different anti-VEGF-A antibodies and  showing control labelling with the secondary antibodies used. You will pay a special attention to the terminology recommended by both reviewers and improve the writing of the manuscript.   

We would appreciate receiving your revised manuscript by Jun 12 2020 11:59PM. To enhance the reproducibility of your results, we recommend that if applicable you deposit your laboratory protocols in protocols.io, where a protocol can be assigned its own identifier (DOI) such that it can be cited independently in the future. For instructions see: http://journals.plos.org/plosone/s/submission-guidelines#loc-laboratory-protocols

We look forward to receiving your revised manuscript.

Kind regards,

Jean-Léon Thomas

Academic Editor

PLOS ONE

Journal Requirements:

2. In your Methods section, please provide additional details regarding the dogs used in your study and ensure you have described the source. For more information regarding PLOS' policy on materials sharing and reporting, see " ext-link-type="uri" xlink:type="simple">https://journals.plos.org/plosone/s/materials-and-software-sharing#loc-sharing-materials."

3. In your Methods section, please provide additional details regarding participant consent from the owners of the animals. In the ethics statement in the Methods and online submission information, please ensure that you have specified (a) whether consent was informed and (b) what type you obtained (for instance, written or verbal). If the need for consent was waived by the ethics committee, please include this information.

5. Please upload a copy of Figure 5, to which you refer in your text on line 315. If the figure is no longer to be included as part of the submission please remove all reference to it within the text.

Reviewers' comments:

Reviewer's Responses to Questions

**Comments to the Author**

1. Is the manuscript technically sound, and do the data support the conclusions?

Reviewer #1: No

Reviewer #2: Yes

2. Has the statistical analysis been performed appropriately and rigorously? 

Reviewer #1: No

Reviewer #2: Yes

3. Have the authors made all data underlying the findings in their manuscript fully available?

Reviewer #1: Yes

Reviewer #2: Yes

4. Is the manuscript presented in an intelligible fashion and written in standard English?

Reviewer #1: No

Reviewer #2: No

5. Review Comments to the Author

Reviewer #1: This manuscript describes the morphological, ultrasonographic appearance of the canine prostate gland, associated with VEGF protein and gene expression in dogs with prostatic hyperplasia treated with orchiectomy or finasteride. It is an interesting manuscript; however, this reviewer has several concerns regarding the study design and results presentation. Please, see my major and minor comments below.

Major concerns

1. This is one of the major concerns regarding this manuscript. Authors show in the figure cytoplasmic VEGF expression in the prostatic epithelial cells and this finding it is not expected. VEGF-A expression should not be present in epithelial prostatic cells. As authors can confirm in Palmieri et. Al. (2014) (https://doi.org/10.1177/0300985814549951), VEGF-A express only internal blood vessel and not epithelial cells. For human, is the same (https://www.proteinatlas.org/ENSG00000112715-VEGFA/tissue). Unfortunately, in this reviewer opinion, all VEGF-A results are just unspecific binding. Please, check the previous literature to confirm that the VEGF-A expression was misinterpreted in this manuscript.

2. It is widely accepted that canine prostatic hyperplasia (HP) is the most common prostatic disease in intact dogs and authors highlighted this point in introduction section. Thus, using only 15-affected dogs (5 in each group) it is a lower number of subjects. Have authors performed a power analysis? In this reviewer opinion, since PH is a common and the procedure performed are not invasive, authors should include more subjects in each group.

3. Benign prostatic hyperplasia is a redundant term in pathology and in both humans and dogs this term is gradually replaced by Prostatic Hyperplasia. Recently, a group of international veterinary pathologists and clinicians covered by the Oncology Pathology Working Group, have proposed the current Histopathological Terminology Standards for the Reporting of Prostatic Epithelial Lesions in Dogs (Palmieri et al. 2019, doi: 10.1016/j.jcpa.2019.07.005). Thus, it is strongly suggested to use the standardized terminology.

4. A very similar research performed by this own research group (Angrimani et al. 2018, doi: 10.1016/j.theriogenology.2018.03.031) was previously published in Theriogenology. The authors had similar groups. In the previously published paper, the authors included a group of heath dogs without PH clinical signs. Why authors omitted their own similar research work? Are the same subjects used in both studies?

5. This study is lacking a group of Heath dogs. It will be interesting to include a group of heath dogs and a power analysis to confirm that five dogs in each group is statistically significant.

6. The tissue biopsy was performed at the diagnosis ang 60 days treatment? It is unclear how PH was diagnosed. Was biopsy performed or only ultrasound examination?

7. Histological PH analysis. It will be interesting to morphologically compare tissue samples at the diagnosis and 60 days after treatment. Besides that, considerer changing the reference Palmieri, et al. [29] for Palmieri et al. 2019 (doi: 10.1016/j.jcpa.2019.07.005).

8. There is no explanation about the tissue biopsy procedure. Please, clarify how biopsy was performed including, procedures and anatomical site of the tissue biopsy.

9. Immunohistochemistry – the positive and negative controls provided are not a real control. Omitting primary antibody is not a negative control. Why mammary gland was used a positive control do VEGF-A? Since mammary gland is a epithelia tissue, is not expected to express VEGF-A. How about the cross-reactivity of the primary antibody with canine tissue.

10. Results, lines 309-313 – Histology, how authors can confirm homogeneity during biopsy procedure? The results section is lacking correlation among the results.

11. The discussion section is poorly presented with focus on hormonal and ultrasonographic finding and authors perform confrontation of their results with the previously literature seems more literature review than a discussion section. Authors should explain their results, providing convincing hypotheses for the results, instead just confront their results with the previous literature.

Minor point.

1. Standardize description of VEGF-A gene and protein. Usually, gene are writing using italic letters. Abstract, lines 37-38 – “VEGF qPCR 37 expression was”, is not VEGF-A qPCR expression is VEGF-A gene expression or just VEGF-A expression

2. Results section should be organized using subheadings.

3. In conclusion section, the conclusions are based mainly in VEGF-A expression. However, in the whole manuscript, the VEGF-A expression is poorly presented.

4. The English should be revised for spelling and grammar errors.

5. The figure captions are poorly described. The Fig 4 present immunohistochemical images without proper caption. The Fig 4 caption says “Figure 4. Testosterone (A) and dihydrotestosterone (B) concentration in Untreated, Finasteride treated, and Orchiectomy treated groups during the experimental period. * indicate: Contrast – Comparison between Untreated vs. Treatments (Finasteride and 663 Orchiectomy). † indicate: Contrast 2 – Comparison between Finasteride vs. Orchiectomy treated groups. * indicate p”. However, the figure shows Fig 4, A, B, C and D. i.e. the figure 4c seems only tissue stroma (poorly representative biopsy). However, there is no caption and it is uncertain the arrow in this image.

Reviewer #2: Canine prostatic hyperplasia is considered since decades a good animal model for the study of the human counterpart. Several papers focusing BPH are present in literature and the manuscript PONE-D-20-09715 “Benign Prostatic Hyperplasia: vascularization, hemodynamic and hormonal analysis of dogs treated with finasteride or orchiectomy” is in the vein.

The study is interesting however, major revision is required before reconsidering for publication.

A revision of the English will contribute to improve the whole manuscript

A general issue: finasteride. No description of finasteride in the abstract nor in the introduction. Finasteride is not considered among keywords. Finasteride is maybe the key of the study and its composition, action, brief story are totally omitted. It has been already employed in dogs? Yes, but it is not highlighted in the introduction.

Hyperplasia or hypertrophy? An old issue: both. However, let use hyperplasia – BPH.

ABSTRACT is long but not incisive.

Page 2 line 22 – 24. Delete “is related…..prostatic cells”

Page 2 line 24. spend a couple of word on finasteride or put in brackets (anti-androgen)

Page 2 line 30. Delete “prostatic”at the beginning of the sentence and add it between “measure” and “volume”

Page to lines 34-35. “Was similar to expect prostatic volume” – unclear

Page 2 lines 43-44. “more marked androgenic decerase and secvere prostate atrophy” – improve English.

Tables. OK

KEYWORDS. delete "carnivore reproduction" and add finasteride

INTRODUCTION

Insert at least couple of sentences on finasteride

Page 3 line 52. …change the last part of the sentence with “the dog as a good animal model for the study of the human counterpart”.

Page 4 line 71. Since “they” reduce

MATERIAL METHODS reports that study was approved by the Bioethics Committee of the School, but it is not known if the dogs enrolled in the study were from kennel or were owned dogs. In this latter case, owner consensus was required anyway. Please clarify and specify in the text – kennel or owned dogs?

Histological evaluation of prostatic fragments must be improved.

Page 7 line 163. Delete “submitted to deparaffinization” and simply write “section were dewaxed”

Page 8 lines 174-176. Source of the secondary antibody employed in immunohistochemistry.

RESULTS

Page 13 line 303. After 30 days from

Page 13 lines 309-313. Histologic description is poor. Please improve. Did you see cystic acini? Stasis of glandular secretion? Epithelial layer/s of the acini: one, more? Papillary growth in the lumen? Histologic differences between untreated and finasteride groups?

Did you observe inflammatory cells in untreated group?

DISCUSSION is long: shortening is required - - half a page at least.

Issue: nobody knows if untreated and finasteride prostates were different at the beginning of the study, because there is not a time 0 biopsy. It is sure in my opinion that finasteride had a positive effect, however I suggest to remark that you had no idea of the entity of hyperplasia in single cases before tratment.

Page 15 line 349. In corroboration – According to….

TABLES FIGURES. OK

6. PLOS authors have the option to publish the peer review history of their article (what does this mean?). If published, this will include your full peer review and any attached files.

Reviewer #1: No

Reviewer #2: No

---

## [Author Response · Author response to Decision Letter 0]

16 May 2020

Jean-Léon Thomas

Editor-in-Chief 

PLOS-ONE

March 15, 2020

Dear Dr. Jean-Léon Thomas, 

We want to thank the opportunity of submitting our manuscript to this prestigious journal and greatly appreciate all the efforts spent on reviewing our work. We are certain that your contributions greatly improved the quality of this manuscript. We hope our modifications make it suitable for acceptance. 

We have accepted all the reviewers’s suggestions. Some interesting points were raised, which certainly make the text clearer and easier to understand.

Ms. N. PONE-D-20-09715- "Benign Prostatic Hyperplasia: vascularization, hemodynamic and hormonal analysis of dogs treated with finasteride or orchiectomy"

Reviewer #1: The suggestions of this reviewer have been addressed and written in BLUE in the revised manuscript.

Authors show in the figure cytoplasmic VEGF expression in the prostatic epithelial cells and this finding it is not expected. VEGF-A expression should not be present in epithelial prostatic cells. As authors can confirm in Palmieri et. al. (2014) (https://doi.org/10.1177/0300985814549951), VEGF-A express only internal blood vessel and not epithelial cells. For human, is the same (https://www.proteinatlas.org/ENSG00000112715-VEGFA/tissue). Unfortunately, in this reviewer opinion, all VEGF-A results are just unspecific binding. Please, check the previous literature to confirm that the VEGF-A expression was misinterpreted in this manuscript.

We agree with the reviewer on the observation of VEGF immunolocalization. In fact, Palmieri et al. (2014) described the lack of VEGF expression in epithelial cells. However, Shidaifat et al., 2005 (doi: 10.1507/endocrj.k07-009) and Chevalier et al., 2002 (doi: 10.1016/s0303-7207(01)00728-6) also showed VEGF expression in stromal and epithelial compartments. Furthermore, Walsh et al., 2002 (doi: 10.1038/sj.pcan.4500575) observed that 50% of PH cases had positive score for VEGF in epithelial cells. Thus, in order to clear such information, the discussion section was modified accordingly in Lines 400-401.

It is widely accepted that canine prostatic hyperplasia (HP) is the most common prostatic disease in intact dogs and authors highlighted this point in introduction section. Thus, using only 15-affected dogs (5 in each group) it is a lower number of subjects. Have authors performed a power analysis? In this reviewer opinion, since PH is a common and the procedure performed are not invasive, authors should include more subjects in each group.

 In fact, the number of subjects was our concern and we fully agree with the reviewer. Thus, before finalizing the experiment, we performed a power analysis by SAS Power and Sample Size 12 (SAS Institute Inc., Cary, NC, USA), in order to assure the appropriate sample size. We obtained a power of 0.99, which is considered an acceptable statistical power (at least 0.8). More information on the experimental design and sample size was provided in Lines 103-107.

Benign prostatic hyperplasia is a redundant term in pathology and in both humans and dogs this term is gradually replaced by Prostatic Hyperplasia. Recently, a group of international veterinary pathologists and clinicians covered by the Oncology Pathology Working Group, have proposed the current Histopathological Terminology Standards for the Reporting of Prostatic Epithelial Lesions in Dogs (Palmieri et al. 2019, doi: 10.1016/j.jcpa.2019.07.005). Thus, it is strongly suggested to use the standardized terminology.

We agree with the reviewer and the terminology was modified throughout.

A very similar research performed by this own research group (Angrimani et al. 2018, doi: 10.1016/j.theriogenology.2018.03.031) was previously published in Theriogenology. The authors had similar groups. In the previously published paper, the authors included a group of heath dogs without PH clinical signs. Why authors omitted their own similar research work? Are the same subjects used in both studies?

We have been researching PH over the years and several distinct data were obtained. The manuscript referred by the reviewer (Angrimani et al. 2018, doi: 10.1016/j.theriogenology.2018.03.031) had the main goal of analyzing the effect of PH itself (and also finasteride) on prostate vascularization, therefore a non-PH group was necessary. On the other hand, the present manuscript has the objective of comparing two modalities of PH treatment, which does not require a non-PH group in the experimental design. In addition, the subjects of both experiments were different. However, we agree with the reviewer and used such information in Lines 78-79. 

This study is lacking a group of Heath dogs. It will be interesting to include a group of heath dogs and a power analysis to confirm that five dogs in each group is statistically significant.

The present manuscript has the main objective of comparing two modalities of PH treatment, thus, we believe that the experimental design does not require a non-PH group. Additionally, we were ethically concerned of performing a needless prostatic biopsy in healthy dogs, under the risk of internal hemorrhage and urethra lesions. However, we partially agree with the reviewer and with this in mind we opt to analyze our data as an orthogonal contrast, in which a combination of groups increases the number of subjects per group and allow for obtaining estimates of main, nested and interaction effects. Such explanation was included in Lines 265-269.

The tissue biopsy was performed at the diagnosis ang 60 days treatment? It is unclear how PH was diagnosed. Was biopsy performed or only ultrasound examination?

We agree with the reviewer for the lack of methodological information. Therefore, we modified and included more information in Lines 95-97 (presumptive PH diagnosis) and Lines 111-118 (prostate biopsy timing and technique). 

Histological PH analysis. It will be interesting to morphologically compare tissue samples at the diagnosis and 60 days after treatment. Besides that, considerer changing the reference Palmieri, et al. [29] for Palmieri et al. 2019 (doi: 10.1016/j.jcpa.2019.07.005).

We agree with the reviewer. However, we opt to perform a single prostate biopsy after 60 days of treatment for ethical reasons (two procedures within a two-month interval) of owned dogs. The indicated reference was included accordingly in Line 180.

There is no explanation about the tissue biopsy procedure. Please, clarify how biopsy was performed including, procedures and anatomical site of the tissue biopsy.

We apologize for the lack of information. Thus, more information on this subject was included in Lines 111-118.

Immunohistochemistry – the positive and negative controls provided are not a real control. Omitting primary antibody is not a negative control. Why mammary gland was used a positive control do VEGF-A? Since mammary gland is a epithelia tissue, is not expected to express VEGF-A. How about the cross-reactivity of the primary antibody with canine tissue.

We apologize for the lack of information. Thus, more information on this subject was included in Lines 197-201.

Results, lines 309-313 – Histology, how authors can confirm homogeneity during biopsy procedure? The results section is lacking correlation among the results.

We apologize for the lack of information. Thus, more information on this subject was included in Lines 180-184 and 344-346. 

The discussion section is poorly presented with focus on hormonal and ultrasonographic finding and authors perform confrontation of their results with the previously literature seems more literature review than a discussion section. Authors should explain their results, providing convincing hypotheses for the results, instead just confront their results with the previous literature.

Discussion was fully revised and modified.

Standardize description of VEGF-A gene and protein. Usually, gene are writing using italic letters. Abstract, lines 37-38 – “VEGF qPCR expression was”, is not VEGF-A qPCR expression is VEGF-A gene expression or just VEGF-A expression.

It was modified accordingly throughout.

Results section should be organized using subheadings.

It was modified accordingly.

Minor Concerns: In conclusion section, the conclusions are based mainly in VEGF-A expression. However, in the whole manuscript, the VEGF-A expression is poorly presented.

We agree with the reviewer and the Conclusion was modified accordingly in Lines 449-453. 

Minor Concerns: The English should be revised for spelling and grammar errors.

The manuscript was submitted to the correction of a native English speaker and all the grammar and spelling errors were corrected throughout. We are sending attached a translation certificate. 

The figure captions are poorly described. The Fig 4 present immunohistochemical images without proper caption. However, the figure shows Fig 4, A, B, C and D. i.e. the figure 4c seems only tissue stroma (poorly representative biopsy). However, there is no caption and it is uncertain the arrow in this image.

We apologize for our mistake and, in fact, figure captions were exchanged. All figure captions were reviewed and modified accordingly. 

We have to clarify that only tissue stroma was observed in our biopsy analysis of the Orchiectomy Treated Group. Thus, in order to clear such query, we added this information in Lines 344-346.

Reviewer #2: The suggestions of this reviewer have been addressed and written in RED in the revised manuscript.

Major Concerns: A revision of the English will contribute to improve the whole manuscript

The manuscript was submitted to the correction of a native English speaker and all the grammar and spelling errors were corrected throughout. We are sending attached a translation certificate. 

A general issue: finasteride. No description of finasteride in the abstract nor in the introduction. Finasteride is not considered among keywords. Finasteride is maybe the key of the study and its composition, action, brief story are totally omitted. It has been already employed in dogs? Yes, but it is not highlighted in the introduction.

We agree with the reviewer and, in fact, this information was missing. Thus, we modified the Abstract and the Introduction section accordingly in Lines 21-24 and Lines 77-81, respectively.

Hyperplasia or hypertrophy? An old issue: both. However, let use hyperplasia – BPH.

We agree with the reviewer that both tissue alterations occur in BPH. However, according to reviewer 1, the novel nomenclature of BPH is Prostatic Hyperplasia, which we adopt throughout. 

Abstract Concerns: ABSTRACT is long but not incisive.

We modified the abstract accordingly.

Introduction Concerns: Insert at least couple of sentences on finasteride

In fact, information on finasteride was missing. We included more information in Lines 77-81.

MATERIAL METHODS reports that study was approved by the Bioethics Committee of the School, but it is not known if the dogs enrolled in the study were from kennel or were owned dogs. In this latter case, owner consensus was required anyway. Please clarify and specify in the text – kennel or owned dogs?

We apologize for the lack of information, which was provided in Lines 92-93.

Material Methods Concerns: Histological evaluation of prostatic fragments must be improved.

More information was included in Lines 180-184.

Material Methods Concerns: Page 8 lines 174-176. Source of the secondary antibody employed in immunohistochemistry.

More information was included in Lines 193-196.

Results Concerns: Page 13 lines 309-313. Histologic description is poor. Please improve. Did you see cystic acini? Stasis of glandular secretion? Epithelial layer/s of the acini: one, more? Papillary growth in the lumen? Histologic differences between untreated and finasteride groups? Did you observe inflammatory cells in untreated group?

More information was included in Lines 339-343.

DISCUSSION is long: shortening is required - - half a page at least.

We agree with the reviewer, therefore the discussion was revised.

Issue: nobody knows if untreated and finasteride prostates were different at the beginning of the study, because there is not a time 0 biopsy. It is sure in my opinion that finasteride had a positive effect, however I suggest to remark that you had no idea of the entity of hyperplasia in single cases before tratment.

We agree with the reviewer. However, we opt to perform a single prostate biopsy after 60 days of treatment for ethical reasons (two procedures within a two-month interval) of owned dogs. More information was included accordingly in Lines 405-409.

We are submitting the revised manuscript to be revaluated for publication. 

Sincerely,

Camila Infantosi Vannucchi, PhD

Professor, Department of Animal Reproduction - University of São Paulo

---

## [Decision Letter · Decision Letter 1]

2 Jun 2020

Prostatic Hyperplasia: vascularization, hemodynamic and hormonal analysis of dogs treated with finasteride or orchiectomy

PONE-D-20-09715R1

Dear Dr. Vannucchi,

We are pleased to inform you that your manuscript has been judged scientifically suitable for publication and will be formally accepted for publication once it complies with all outstanding technical requirements.

With kind regards,

Jean-Léon Thomas

Academic Editor

PLOS ONE

Additional Editor Comments (optional):

Reviewers' comments:

Reviewer's Responses to Questions

**Comments to the Author**

1. If the authors have adequately addressed your comments raised in a previous round of review and you feel that this manuscript is now acceptable for publication, you may indicate that here to bypass the “Comments to the Author” section, enter your conflict of interest statement in the “Confidential to Editor” section, and submit your "Accept" recommendation.

Reviewer #1: All comments have been addressed

Reviewer #2: All comments have been addressed

2. Is the manuscript technically sound, and do the data support the conclusions?

Reviewer #1: Yes

Reviewer #2: Yes

3. Has the statistical analysis been performed appropriately and rigorously? 

Reviewer #1: Yes

Reviewer #2: Yes

4. Have the authors made all data underlying the findings in their manuscript fully available?

Reviewer #1: Yes

Reviewer #2: Yes

5. Is the manuscript presented in an intelligible fashion and written in standard English?

Reviewer #1: Yes

Reviewer #2: Yes

6. Review Comments to the Author

Reviewer #1: The authors provided a revised manuscript with all concerns properly addressed. After reviewing the revised version of this manuscript, I have no further comments and I recommend the revised version of this manuscript for publication in PLos One.

Stay safe.

Reviewer #2: In the second version of the manuscript all suggestions given have been considered. In particular, Finasteride was better described, hyperplasia instead of hypertrophy was used, an improvement of Materials and Methods was performed. Discussion was also revised.

The manuscript, in my opinion, is now suitable for publication.

7. PLOS authors have the option to publish the peer review history of their article (what does this mean?). If published, this will include your full peer review and any attached files.

Reviewer #1: No

Reviewer #2: No

---

## [Editor Report · Acceptance letter]

16 Jun 2020

PONE-D-20-09715R1 

Prostatic Hyperplasia: vascularization, hemodynamic and hormonal analysis of dogs treated with finasteride or orchiectomy 

Dear Dr. Vannucchi:

I'm pleased to inform you that your manuscript has been deemed suitable for publication in PLOS ONE. Congratulations! Your manuscript is now with our production department. 

Kind regards, 

on behalf of

Dr. Jean-Léon Thomas 

Academic Editor

PLOS ONE